# Inhibition of IRAK1 Is an Effective Therapy for Autoimmune Hypophysitis in Mice

**DOI:** 10.3390/ijms232314958

**Published:** 2022-11-29

**Authors:** Hsiao-Chen Huang, Yun-Ti Chen, Han-Huei Lin, Zhi-Qin Li, Jinn-Moon Yang, Shey-Cherng Tzou

**Affiliations:** 1Department of Biological Science and Technology, National Yang Ming Chiao Tung University, Hsinchu 300, Taiwan; 2Institute of Bioinformatics and Systems Biology, National Yang Ming Chiao Tung University, Hsinchu 300, Taiwan; 3Institute of Molecular Medicine and Bioengineering, National Yang Ming Chiao Tung University, Hsinchu 300, Taiwan; 4Center for Intelligent Drug Systems and Smart Bio-Devices, National Yang Ming Chiao Tung University, Hsinchu 300, Taiwan

**Keywords:** autoimmune hypophysitis, interleukin-1 receptor-associated kinase 1 (IRAK1), inhibitor, pathogenesis

## Abstract

Autoimmune hypophysitis (AH) is an autoimmune disease of the pituitary for which the pathogenesis is incompletely known. AH is often treated with corticosteroids; however, steroids may lead to considerable side effects. Using a mouse model of AH (experimental autoimmune hypophysitis, EAH), we show that interleukin-1 receptor-associated kinase 1 (IRAK1) is upregulated in the pituitaries of mice that developed EAH. We identified rosoxacin as a specific inhibitor for IRAK1 and found it could treat EAH. Rosoxacin treatment at an early stage (day 0–13) slightly reduced disease severity, whereas treatment at a later stage (day 14–27) significantly suppressed EAH. Further investigation indicated rosoxacin reduced production of autoantigen-specific antibodies. Rosoxacin downregulated production of cytokines and chemokines that may dampen T cell differentiation or recruitment to the pituitary. Finally, rosoxacin downregulated class II major histocompatibility complex expression on antigen-presenting cells that may lead to impaired activation of autoantigen-specific T cells. These data suggest that IRAK1 may play a pathogenic role in AH and that rosoxacin may be an effective drug for AH and other inflammatory diseases involving IRAK1 dysregulation.

## 1. Introduction

Autoimmune hypophysitis (AH) is an inflammatory disease caused by the infiltration of autoreactive lymphocytes and other immune cells into the pituitary [1]. Infiltration of the immune cells and inflammation initially lead to pituitary expansion that causes headache and visual disturbance. As AH progresses, destruction of endocrine cells and hormonal impairments (hypopituitarism) frequently occur [1,2]. Prompt medical care is vital because AH can be fatal if neglected [3,4], although fatal cases are less often seen due to increased awareness of the disease. AH patients are often managed by corticosteroids [5]; however, long-term use of steroids may lead to considerable side effects [6]. AH may reoccur after tapering and discontinuation of corticosteroids [7]. Currently, no drug that targets specific pathogenic mechanisms in AH is available. The development of safe and effective therapy for AH is desirable.

Although AH was considered a rare disease of the pituitary gland [1,8], novel cancer immunotherapy using immune checkpoint inhibitors has been reported to cause AH. For example, ipilimumab, an anti-CTLA-4 antibody approved by FDA for treating cancers, induced a surprisingly high incidence of AH (4.5–10%) in cancer patients [9,10,11]. Similarly, monoclonal antibodies to programmed cell death protein-1 (PD-1)/programmed cell death 1 ligand 1 (PD-L1) can induce AH, albeit less frequently than the CTLA-4 antibody. On the other hand, patients receiving type I interferons for the treatment of chronic hepatitis C virus (HCV) infection may develop AH-like symptoms [12,13,14]. Although these immune-boosting treatments may be beneficial to treat specific diseases, they may bear the cost of immune-related adverse effects such as AH. These findings highlight a pressing need to understand the pathogenesis of AH and to better manage AH induced by immunotherapies.

Pathogen- or danger-associated molecular patterns (PAMPs/DAMPs) and the corresponding pattern-recognition receptors (PRRs) are involved in the pathogeneses of several autoimmune diseases [15,16]. For example, toll-like receptor (TLR) 7 signaling promotes autoantibody production in lupus [17,18], whereas TLR2/4 contribute to rheumatoid arthritis [19,20], and TLR2/4/9 contribute to multiple sclerosis [21] pathogeneses. It seems rational that blocking the PRRs may be helpful for these inflammatory diseases. In this regard, the interleukin-1 receptor-associated kinases (IRAKs) may represent promising candidates for drug development as they are important signaling hubs downstream of several TLRs. However, whether PRR (TLR) signaling promotes AH progression and whether inhibition of the IRAK1-mediated signaling pathway could treat AH are currently unknown. Furthermore, few inhibitors are currently available to inhibit aberrant IRAK1 activity in diseases.

To further investigate the pathogenesis and develop a new therapy for AH, we tested whether a novel IRAK1 inhibitor identified in our lab, rosoxacin, could treat a mouse model of AH (experimental autoimmune hypophysitis, EAH) that we previously established [22,23]. We compared IRAK1 expressions in the pituitaries of non-diseased and diseased mice. We next tested the efficacy of rosoxacin in ameliorating EAH. Finally, we explored the potential mechanisms of rosoxacin treatment by in vitro assays. Our results indicate rosoxacin is an effective treatment for EAH and may be worthwhile to further evaluate in human AH patients.

## 2. Results

### 2.1. Higher IRAK1 Was Expressed in the Pituitaries of Mice That Developed EAH

Female SJL/J mice developed florid EAH on day 28 post-immunization with mouse growth hormone (mGH) [23]. In contrast, mice immunized by CFA alone did not show any sign of lymphocytic infiltration in the pituitaries. To explore whether IRAK1 may be involved in the pathogenesis of EAH, we first assessed protein expression in the pooled pituitaries of CFA-immunized mice and mGH-immunized mice by western blotting. A basal level of IRAK1 was expressed in the pituitaries of non-diseased (CFA-immunized) mice. IRAK1 expression in the pituitaries increased by 42.8% (normalized IRAK1 expression in CFA-immunized mice: 0.28; normalized IRAK1 expression in mGH-immunized mice: 0.40) when mice developed EAH, following immunization with mGH (Figure 1A). IRAK1 expression in the inguinal lymph nodes that drain the immunization sites did not differ (normalized IRAK1 expression in CFA-immunized mice: 0.18; normalized IRAK1 expression in mGH-immunized mice: 0.19) (Figure 1B). This result suggests that inflammation in the pituitaries led to upregulation of IRAK1 in situ.

### 2.2. Rosoxacin Suppressed EAH

To identify an inhibitor for IRAK1, we docked 2122 FDA drugs obtained from the Protein Data Bank [24] to the binding site of IRAK1 (PDB ID: 6BFN) using an in-house developed software GEMDOCK [25]. Rosoxacin was the top candidate from the prediction, given 10 μM of rosoxacin inhibited ~50% IRAK1 activity, compared to less than 5% for other kinases tested (manuscript in preparation). Furthermore, rosoxacin showed a > 50-fold selectivity for IRAK1 over IRAK4, which is a kinase highly homologous to IRAK1. These data indicate that rosoxacin is a potential drug to inhibit IRAK1.

TLRs are involved in the pathogeneses of autoimmune diseases. Since IRAK1 is a central signaling molecule in IL-1 and TLR pathways, and its expression was upregulated in the pituitaries of mice that developed EAH, we tested whether inhibition of IRAK1 could suppress EAH progression. Female SJL/J mice were immunized with mGH and treated with PBS or rosoxacin. Histological examination confirmed that PBS-treated mice developed florid EAH after immunization with mGH. Numerous lymphocytes infiltrated the pituitary, sometimes forming large aggregates within the parenchyma (Figure 2A). Oral administration of rosoxacin on day 0–13 (early treatment) seemed to reduce EAH slightly in some mice; however, this effect was not significant (*p* = 0.376, PBS vs. rosoxacin day 0–13). In contrast, oral gavage of rosoxacin on day 14–27 post-immunization (late treatment) significantly reduced EAH severity in mGH-immunized mice (*p* = 0.046, PBS vs. rosoxacin day 14–27) (Figure 2A). Interestingly, we observed that a small number of lymphocytes was confined in the sinusoidal spaces but not in the parenchyma of the pituitaries of mice that received late rosoxacin treatments.

T cells constitute the primary cell type among the infiltrating cells; therefore, we assessed whether rosoxacin reduced T cell infiltration into the pituitaries of mice that developed EAH. Immunohistochemical staining using an anti-CD3 antibody revealed the most intensive T cell infiltration occurred in the pituitaries of PBS-treated mice (Figure 2B). In contrast, only a few T cells were identified in the pituitary sections of mice treated with rosoxacin on day 14–27. Treatment of rosoxacin on day 0–13 reduced T cell infiltration but was not as effective as the late treatment. Thus, these results suggest that rosoxacin may be an effective treatment for EAH.

### 2.3. Rosoxacin Inhibited Autoantibody Production

Although EAH is a T cell-mediated disease, autoantibodies to pituitary antigens could correlate with disease induction [22]. We examined whether rosoxacin treatment could alter autoantibody production in vivo and in vitro. We first analyzed mGH antibody titers in the sera of PBS-treated or rosoxacin-treated mice. We found that both early and late rosoxacin treatments show a trend to reduce the autoantibodies in the treated mice. Interestingly, early rosoxacin treatments more potently inhibited the production of mGH antibodies than did late rosoxacin treatments (Figure 3A). We next examined the in vitro production of mGH antibodies by lymphocytes isolated from deep cervical lymph nodes that drain the pituitary [26,27]. We found that lymphocytes isolated from rosoxacin-treated mice produced significantly fewer mGH antibodies in vitro. Similar to the mGH antibody titers in vivo, early rosoxacin treatments more potently inhibited mGH antibody production by isolated lymphocytes in vitro (Figure 3B). We also examined the in vitro production of mGH antibodies by lymphocytes isolated from inguinal lymph nodes that drained the immunization sites. Lymphocytes isolated from PBS-treated mice produced high amounts of mGH antibodies in culture. Adding rosoxacin to the lymphocyte culture for 72 h marginally reduced the production of mGH antibodies (Figure 3C). In contrast, adding rosoxacin to the lymphocytes isolated from rosoxacin-treated mice significantly inhibited the production of the antibodies (Figure 3C). We conclude that rosoxacin reduces autoantibody production by the B cells in vitro and in vivo.

### 2.4. Rosoxacin Downregulated Production of Cytokines and Chemokines In Vitro

Cytokines and chemokines are key signaling molecules that orchestrate cell functions in the immune system. To gain insights into how rosoxacin may treat EAH, we analyzed the production of cytokines and chemokines in vitro by cells isolated from deep cervical lymph nodes. Cell culture supernatants were simultaneously measured for 23 different cytokines and chemokines by a multiplex cytokine array. We found several cytokines and chemokines were downregulated by rosoxacin treatments (Figure 4). In general, late rosoxacin treatments attained a higher degree of downregulation of cytokines and chemokines than did the early treatments, suggesting treatment effects may wane in the early treatment group. In particular, interleukin (IL)-1β (reduced by 2.67-fold and 4.77-fold for early and late treatments, respectively), granulocyte-colony-stimulating factor (G-CSF) (4.9-fold and 10.29-fold), C-X-C motif chemokine ligand 1 (CXCL1) (6.70-fold and 15.71-fold), and CC motif chemokine ligand 3 (CCL3) (3.68-fold and 4.23-fold) were markedly reduced. Although the expression levels of IL-2 (2.29-fold and 3.48-fold), IL-12 (2.83-fold and 7.04-fold), interferon (IFN)-γ (2.61-fold and 2.06-fold), CCL4 (15-fold and 2.37-fold), and CCL5 (2.10-fold and 4.26-fold) were lower than the cytokines/chemokines above, their expression was also downregulated by rosoxacin. Appendix A shows the expression of all the cytokines/chemokines tested in the study.

### 2.5. Rosoxacin Inhibits Class II Major Histocompatibility Complex on Antigen-Presenting Cells In Vitro

The effect of rosoxacin on the production of cytokines and chemokines suggests that rosoxacin may hinder the activation and differentiation of autoreactive lymphocytes in the draining lymph nodes of the pituitary to alleviate EAH. Class II major histocompatibility complex (class II MHC) and CD80 are critical molecules used by antigen-presenting cells to activate T cells. We thus analyzed the effects of rosoxacin on the expressions of these two molecules on a mouse macrophage cell line RAW264.7. Interestingly, rosoxacin reduced the expression of class II MHC but not CD80 on RAW264.7 cells under steady (unstimulated) conditions in vitro (Figure 5). We used lipopolysaccharide (LPS) to induce an inflammatory state in RAW264.7 cells, and we found that LPS upregulated the expressions of both class II MHC and CD80. Adding rosoxacin to RAW264.7 cells dampened the expression of class II MHC (but not CD80) induced by LPS, indicating that rosoxacin may block antigen presentation to T cells.

## 3. Discussion

Due to increased awareness and advances in radiological imaging, AH patients are less often misdiagnosed and undergo operation [11]. Instead, steroids and immunosuppressants are used as first-line treatments for AH. Drugs that target key components in the disease mechanisms are unavailable in AH thus far due to a lack of understanding of its pathogenesis. Therefore, the development of drugs that inhibit key pathogenic mediators could more effectively treat AH. Our study demonstrates that rosoxacin, an IRAK1-specific inhibitor identified in our lab, can suppress EAH. Our study also indicates that inhibition of IRAK1 may interfere with the activation and differentiation of autoreactive lymphocytes in the draining lymph node or inhibit the infiltration of activated lymphocytes in the pituitary. Therefore, our study not only proposes a useful treatment but also adds new insights into the pathogenesis of EAH. More studies are necessary to uncover the detailed mechanisms of treatment effects of rosoxacin.

Rosoxacin is one of the first-generation quinolone derivative antibiotics for treating bacterial infections. In this study, we repurposed rosoxacin for EAH treatment. Although intestinal microbiota is linked to autoimmune diseases [28], the treatment effects of rosoxacin on EAH cannot be solely explained by the alteration of intestinal microbiota. First, rosoxacin inhibited IRAK1, specifically, but not the related IRAK4 and other kinases. Second, rosoxacin inhibited antibody production by isolated lymphocytes in vitro. Finally, rosoxacin inhibited class II MHC, but not CD80, expression by mouse Raw264.7 macrophage cells. These in vitro cell culture experiments were carefully conducted under sterile conditions. These data suggest that rosoxacin may exert a direct treatment effect on EAH by inhibiting IRAK1. Since IRAK1 is a critical signaling molecule downstream of IL-1R and many TLRs, and inhibitors for IRAK1 are relatively scarce, rosoxacin may be further developed into a valuable drug for treating diseases associated with aberrant activation of various TLR pathways, such as autoimmune diseases and cancers [29].

To understand the pathogenesis of AH, we had previously established a mouse model that highly mimics the human disease [22,30]. We subsequently identified that growth hormone is a pathogenic autoantigen in this model and demonstrated that autoreactive T cells received a second activation within the pituitary [23]. Following the second activation, the pituitary-infiltrating T cells proliferated in situ in the pituitary and secreted high amounts of IFN-γ and IL-17. In the current study, we provide evidence that IRAK1 may contribute to the development of EAH. Intracellular DAMPs (such as nuclear proteins/nucleic acids) released from dying or dead pituitary cells can activate antigen-presenting cells by TLR-IRAK pathways to enhance presentation of autoantigens to pituitary-infiltrating T cells, leading to activation and proliferation of the T cells to aggravate disease progression. Thus, our findings that rosoxacin can treat EAH suggests IL-1R/TLRs-IRAK1 play an important role in EAH pathogenesis. Further studies are needed to elucidate the identity of the specific TLRs involved.

IL-1β expression in the deep cervical lymph nodes of the mGH-immunized mice was downregulated by rosoxacin. Interestingly, IL-1β is linked to the differentiation of Th17 cells [31,32]. Thus, it is rational to speculate that inhibiting IL-1β production by rosoxacin in the lymph nodes may hamper the differentiation of functional Th17 cells that later infiltrate the pituitary. On the other hand, IL-1β has been demonstrated to upregulate the endothelial expression of intercellular adhesion molecule-1 (ICAM-1) and vascular cell adhesion molecule-1 (VCAM-1) [33,34,35], which are crucial for recruiting circulating lymphocytes into the inflammatory tissue [36]. Thus, rosoxacin may suppress EAH development by downregulating IL-1β-induced ICAM-1/VCAM-1 expression on the endothelial cells. Our in vivo studies are consistent with this notion, as we found lymphocytes were confined to the sinusoidal spaces but not to the pituitary parenchyma of the mice in the late rosoxacin treatment group. Other cytokines/chemokines, such as G-CSF, CXCL1, and CCL3, in the deep cervical lymph nodes were also markedly downregulated by rosoxacin. G-CSF is well-known to promote hematopoiesis in bone marrow [37]. Both G-CSF and CXCL1 are also known to promote neutrophil recruitment and activity [38,39,40]. Interestingly, IL-1β enhances G-CSF secretion to promote hematopoiesis, at least in some pathological conditions [41]. IL-1β also regulates CXCL1 expression [42]. Thus, IL-1β-GCSF-CXCL1 may form an amplifying loop that promotes the generation, recruitment, and activity of neutrophils, despite the precise role of neutrophils in AH/EAH not being well-characterized.

Lymphocytes start to infiltrate the pituitary on ~day 14 post-immunization in EAH [22]. Activation and differentiation of autoreactive lymphocytes most likely occur in the draining lymph node during day 0–13, whereas migration of the cells into the pituitary occurs after day 14. We design treatment schemes using day 14 as a cut-off point, aiming to block the activation/differentiation by early rosoxacin treatment or block EAH progression after T cell infiltration. Early rosoxacin treatments were less effective in suppressing EAH, although they were capable of downregulating antibody and cytokine/chemokine production. The reason for the ineffectiveness is currently unknown; however, we noted that early rosoxacin treatments did not downregulate cytokine/chemokine expression as drastically as did the late rosoxacin treatments. As cytokines modulate T cell differentiation, residual IL-1β expression may be sufficient to sustain T cell differentiation or induce enough cell adhesion molecules on the endothelial cells in the pituitary. Alternatively, late rosoxacin treatment may block the antigen presentation and the second activation and proliferation of the pituitary-infiltrating T cells, as we have previously reported [23].

In conclusion, we show that inhibition of IRAK1 by a specific inhibitor rosoxacin could suppress EAH, possibly by inhibiting the activation and/or differentiation of autoreactive T cells in the pituitary-draining lymph nodes or by blocking the infiltration of T cells into the pituitary. These data may provide a useful therapy and point to further investigations into the role of TLR-IRAK1 in the pathogenesis of AH.

## 4. Materials and Methods

### 4.1. Mice

SJL/J mice were purchased from the Jackson Laboratory (Bar Harbor, ME, USA) and bred in the animal facility of the National Yang Ming Chiao Tung University. Eight- to ten-week-old female SJL/J mice were used for induction of EAH. All experiments were approved and conducted in accordance with the standards established by the National Yang Ming Chiao Tung University Animal Care and Use Committee.

### 4.2. Production and Purification of Recombinant Mouse Growth Hormone (mGH)

Production and purification of recombinant mGH were described in detail previously [23]. In brief, a bacterial expression vector encoding a histidine tag-sumo peptide-mGH fusion protein was transformed into Shuffle T7 Express competent cells (NEB, Ipswich, MA, USA). Production of recombinant histidine tag-sumo peptide-mGH fusion proteins was induced by IPTG (Invitrogen, Carlsbad, CA, USA). The bacterial cells were harvested and lysed, and the histidine tag-sumo peptide-mGH was purified by a nickel-NTA agarose column (GE Life Sciences, Marlborough, MA, USA). Purified histidine tag-sumo peptide-mGH was cleaved with sumo proteases at the junction of sumo peptide and mGH. Histidine tag-sumo peptide was removed from mGH by a nickel-NTA agarose column. Finally, mGH was purified to near homogeneity by an S-100 size-exclusion column (GE Life Sciences). A representative result of mGH purification is shown in Appendix A.

### 4.3. Induction of EAH and Rosoxacin Treatments

Induction of EAH by immunization of mGH was previously reported [43]. Briefly, purified mGH was emulsified 1:1 with complete Freund’s adjuvant (Sigma-Aldrich, St. Louis, MO, USA) supplemented with 4 mg/mL of heat-inactivated M. tuberculosis extracts. Mice were injected with the emulsion (300 μg mGH in 100 μL) subcutaneously into the dorsal hind leg region (50 μL) and the contralateral inguinal region (50 μL). Emulsions were injected again on day 7 in the opposite sites. Mice immunized with CFA only did not develop EAH and were used as the non-diseased control.

Rosoxacin was dissolved in 0.1N NaOH and then pH was adjusted to ~7.5 by PBS. Rosoxacin (200 μg/in 150 μL PBS) was orally administered to EAH mice once daily on days 0–13 for early treatment or on days 14–27 for late treatment. EAH mice that received PBS only were used as a control group. Mice were sacrificed on day 28 post-immunization; sera, pituitaries, deep cervical lymph nodes, and inguinal lymph nodes were collected for downstream analyses.

### 4.4. IRAK1 Expression in Pituitaries and Inguinal Lymph Nodes

Pituitaries (*n* = 6 per group) and inguinal lymph nodes were homogenized in lysis buffer (Abcam, Cambridge, MA, USA) by mortar and pestle in liquid nitrogen. Lysates were centrifuged to remove debris and quantified by a BCA kit (Thermo Fisher Scientific, Waltham, MA, USA). Cleared lysates (60 μg/lane) were separated by a reducing SDS–PAGE and transferred to a nitrocellulose membrane. After blocking with 3% bovine serum albumin in PBST (PBS containing 0.05% Tween-20 (VWR-Amresco, Radnor, PA, USA)), the membranes were incubated with an anti-IRAK1 antibody (1:1000, Cell Signaling, Danvers, MA, USA), followed by a peroxidase-conjugated anti-rabbit secondary antibody (1:20,000, Jackson ImmunoResearch, West Grove, PA, USA). The blots were developed by adding a chemiluminescent substrate (Thermo Scientific), and chemiluminescence images were recorded by a CCD imaging system (Hansor, Taichung, Taiwan). The images of IRAK1 and *β*-actin blots on the same membrane were quantified with the Image J software (version 13.06 for Mac OS X, 64 bit, free software, National Institutes of Health, Bethesda, MD, USA (accessed on 25 August 2021)). Signal intensities of IRAK1 were divided by the intensities of *β*-actin to obtain normalized IRAK1 expression.

### 4.5. Histological Analyses

Pituitaries were harvested on day 28 post-immunization, fixed in 10% PBS-buffered formalin, and embedded in paraffin. The pituitary sections were cut (4 μm) and stained with hematoxylin and eosin for histopathological examination under a light microscope by two independent investigators. Disease scores were given as 0 for no infiltration; 1 for <20% pituitary area infiltrated; 2 for 20–40% pituitary area infiltrated; 3 for 40–60% pituitary area infiltrated; 4 for 60–80% pituitary area infiltrated; 5 for 80–100% pituitary area infiltrated. To characterize the T cell (CD3^+^) infiltrates, we stained pituitary sections of different treatment groups with a CD3 antibody, as described previously [23]. For quantification of pituitary area infiltrated by CD3^+^ T cells, the stained sections were randomly chosen for five fields under a light microscope. The area positively stained for CD3 was selected and calculated by ImageJ, then divided by the total area of the field. The calculated proportion of the five fields were averaged and presented as “CD3-positive area (%)”. 

### 4.6. Determination of mGH Autoantibody Production

To determine mGH autoantibody titer in the sera of mice on day 28 post-immunization, sera were diluted to 1:3000 and added to the ELISA plates precoated with mGH (0.2 μg/well). mGH autoantibodies on the plates were detected by a peroxidase-conjugated anti-mouse IgM + IgA + IgG secondary antibody, followed by a peroxidase substrate TMB (Invitrogen). Color development was stopped by 0.1 N HCl, then measured at 450 nm by a microplate reader (Molecular Device, San Jose, CA, USA). To determine the in vitro production of mGH autoantibody by lymphocytes, we isolated lymphocytes from deep cervical lymph nodes of mice from different treatment groups and cultured the cells in DMEM containing 10% fetal bovine sera (FBS) for 72 h. To test the effect on autoantibody production, rosoxacin was added to lymphocytes isolated from inguinal lymph nodes of mice from different treatment groups for 72 h. Culture supernatants were added to the ELISA plates precoated with mGH, and the mGH autoantibodies were measured as described above.

### 4.7. Multiplex Cytokine Analysis

Deep cervical lymph nodes were harvested from mice from different treatment groups on day 28 post-immunization. Lymphocytes were isolated from the lymph nodes and cultured (2 × 10^5^ cells/200 μL) in a 96-well plate for 96 h. The culture supernatants were collected and analyzed for cytokine and chemokine production by a multiplex cytokine assay (Bio-Plex Multiplex Immunoassay System, Bio-Rad, Hercules, CA, USA). The cytokine array can simultaneously measure 23 cytokines/chemokines, including IL-1*α*, IL-1*β*, IL-2, IL-3, IL-4, IL-5, IL-6, IL-9, IL-10, IL-12 (p40), IL-12 (p70), IL-13, IL-17, G-CSF, GM-CSF, IFN-*γ*, TNF-*α*, Eotaxin, CXCL1, CCL-2, CCL-3, CCL4, and CCL5.

### 4.8. Surface Expression of Class II MHC and CD80 on Raw264.7 Cells

Raw264.7 cells (2 × 10^5^ cells/well) were cultured in DMEM 10% FBS in 24-well plates for 16 h. LPS (1 μg/mL) or PBS were added to the cells for 2 h, then replaced with fresh DMEM 10% FBS containing 2 μM rosoxacin for 48 h. After treatments, the cells were harvested by scrappers, resuspended in DMEM, and stained by FITC-conjugated anti-mouse class II MHC or FITC-conjugated anti-mouse CD80 antibodies. Stained cells were washed and fixed in 2% paraformaldehyde in PBS. Expression of class II MHC and CD80 were analyzed by a flow cytometer (Accuri C6, BD Biosciences, San Jose, CA, USA).

### 4.9. Statistical Analysis

Normalized IRAK1 expression in the pituitaries and the inguinal lymph nodes of mice, disease scores in the pituitaries of different treatment groups, and mGH autoantibody titers were expressed as mean ± SD. The difference in IRAK1 expressions and the effect of rosoxacin on mGH autoantibody production in vitro were assessed by the ranksum test. Kruskal–Wallis test was used to detect differences in disease scores and mGH autoantibodies in sera and in lymphocyte cultures of different treatment groups, followed by the rank-sum test for pairwise comparison.

## Figures and Tables

**Figure 1 ijms-23-14958-f001:**
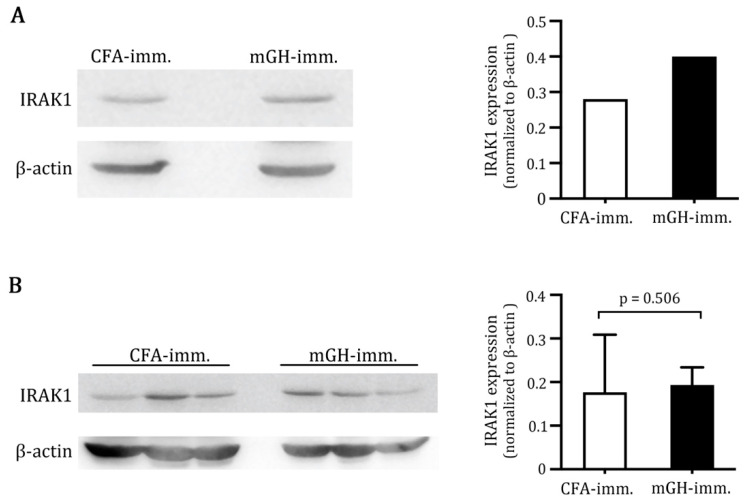
IRAK1 expression in EAH. (**A**) Pituitary lysates of CFA-immunized and mGH-immunized mice were analyzed for IRAK1 expression by western blot using an IRAK1 antibody. (**B**) Inguinal lymph node lysates from CFA-immunized and mGH-immunized mice were analyzed for IRAK1 expression by western blot using an IRAK1 antibody.

**Figure 2 ijms-23-14958-f002:**
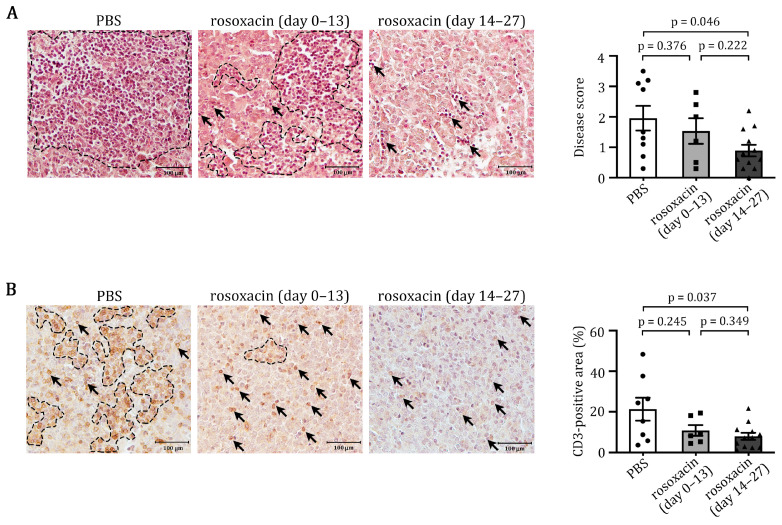
Rosoxacin suppressed EAH development. (**A**) Mice that developed EAH were treated with PBS or rosoxacin at early (day 0–13) or late (day 14–27) stages. Treated mice were sacrificed and the pituitaries were cut and stained with H&E. Infiltrating mononuclear cells in the sections were enclosed by dashed line or indicated by arrows. Scale bar: 100 μm. (**B**) Pituitary sections of the mice were immunostained for CD3 T cells. CD3^+^ T cells in the sections are enclosed by dash-line or indicated by arrows. Each symbol (circles, squares, and triangles) in the histogram plots represents the averaged data of a single mouse. Scale bar: 100 μm.

**Figure 3 ijms-23-14958-f003:**
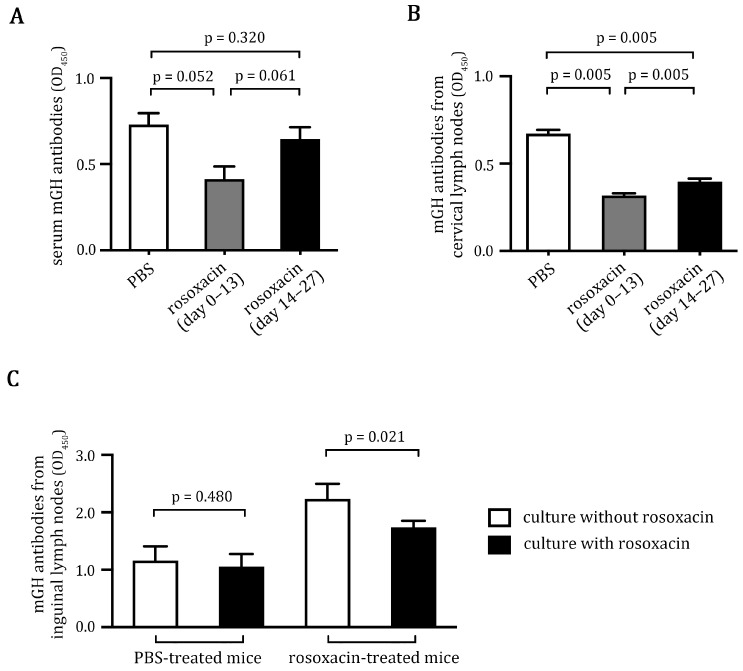
Rosoxacin reduced autoantibody production. (**A**) Anti-mGH antibodies in the sera of EAH mice that received different treatments were determined by an ELISA. (**B**) Lymphocytes isolated from deep cervical lymph nodes of EAH mice that received different treatments were cultured for 72 h. The production of anti-mGH antibodies by the lymphocytes were determined by an ELISA. (**C**) Lymphocytes isolated from inguinal lymph nodes of EAH mice that received different treatments were cultured for 72 h with (filled bars) or without (open bars) rosoxacin. The production of anti-mGH antibodies by the lymphocytes was determined by an ELISA.

**Figure 4 ijms-23-14958-f004:**
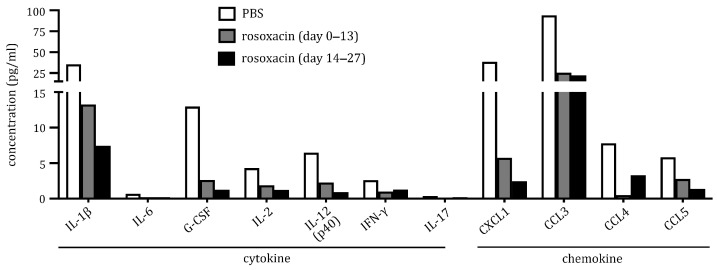
Rosoxacin reduced secretions of cytokines and chemokines from lymphocytes. Lymphocytes isolated from deep cervical lymph nodes of EAH mice that received different treatments were cultured for 96 h. The levels of cytokines and chemokines secreted by lymph node cells were determined by a multiplex cytokine assay. Shown are cytokines/chemokines whose expression levels were modulated by rosoxacin treatments.

**Figure 5 ijms-23-14958-f005:**
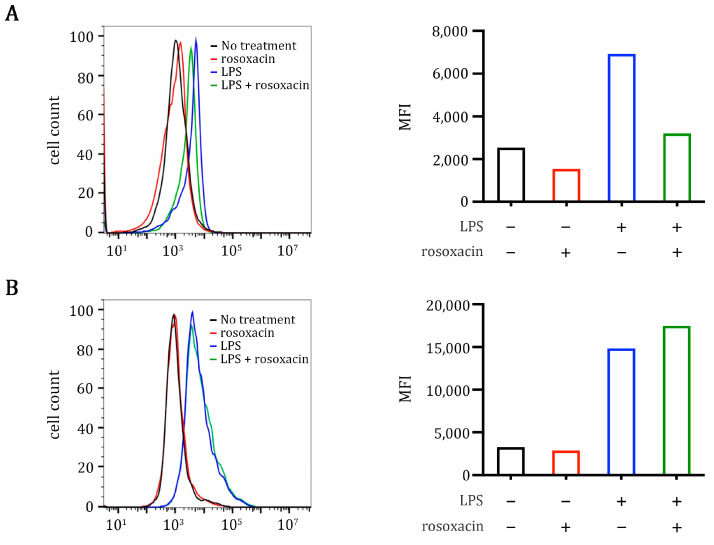
Rosoxacin reduced class II MHC on antigen-presenting cells. RAW 264.7 macrophages were unstimulated (PBS-treated) or stimulated by LPS, followed by rosoxacin treatments. The expressions of class II MHC (**A**) and CD80 (**B**) on the RAW264.7 cells were determined by flow cytometry.

## Data Availability

The data reported in this study are available from the corresponding authors upon reasonable request.

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
