# Peer review of "Inhibition of IRAK1 Is an Effective Therapy for Autoimmune Hypophysitis in Mice"

_ijms, 2022, doi:10.3390/ijms232314958_

Round 1
Reviewer 1 Report
This is a well-written paper on an autoimmune pituitary disease—Lin et al. report on the mechanisms and therapeutic candidates for autoimmune hypophysitis. Not much is known about this disorder; therefore, additional information is welcomed to characterize this entity better. I have a few remarks and suggestions that I believe may improve the quality of the paper.
1) On Page 4, in Figure 2A, did authors obtain data about the pituitary or its peripheral hormone levels other than disease score?
2) On Page 4, in Figure 2B, CD3 positive cells are not unclear because background staining is relatively high. The rosoxacin (day 14-27) group specimen indicated showed other CD3-positive cells in the right upper and right lower sites.
3) In Figure 5, the authors showed that adding rosoxacin to macrophages dampened the expression of class II MHC. Were helper T cells and CD8+ cytotoxic T cells observed in the pituitary? If so, pituitary-specific antigens and specific pituitary injuries are estimated. Still, it contradicts autoimmune hypophysitis causes a variety of clinical features that disturb not limited to particular hormone-producing cells. NK cells are not related to the pathogenesis?
4) Rosoxacin treatment at a late stage suppresses the experimental autoimmune hypophysitis better than at an early stage early stage. Does this phenomenon reflect the clinical feature of autoimmune hypophysitis as a chronic inflammatory disease?
5) In the discussion section, are neutrophils also able to associate with the pathogenesis of autoimmune hypophysitis?
6) To what extent is this model limited by the absence of cytokines such as IL-15 in mice?
Reviewer 2 Report
Lin and colleagues report a very interesting experimental paper about the role of the IRAK1 pathway in a mouse model of autoimmune hypophysitis. The study incorporates both in vivo and in vitro data. In vivo, the authors induced hypophysitis by the injection of a hypophyseal protein (growth hormone), showing that the disease is ameliorated when mice ingest rosoxacin. This is a drug they have identified in silico as capable of inhibiting the activity of IRAK1 by about 50%. Amelioration was assessed via a reduction of the T cell infiltration in the pituitary gland and antibody production. In vitro, the authors demonstrated that rosoxacin reduced the production of several cytokines and chemokines from cultured lymphocytes, and the expression of MHC class II molecules in a macrophage cell line.
The work is clearly written and well referenced. I only have some minor comments/suggestions.
1) Results section. Use past tense for all verbs. For example: “Female SJL/J mice developed florid EAH” rather than “Female SJL/J mice develop florid EAH”.
2) Figure 2A. The figure legend says that panel 2A was stained with hematoxylin and eosin. Hematoxylin stains the nuclei in dark blue, but I have hard time to see a blue staining in the image, which is predominantly red.
3) Figure 3 can be improved. In panel A, the label of the Y-axis is clearer if it says “Serum mGH antibobies”. Similarly, in panel B the Y-label can be made improved ore clear if it says “mGH antibodies from cervical lymph nodes”. Then, panel C is confusing because it is split into two panels, both under the C heading. These two C panels can be combined into one, with a Y-label called “mGH antibodies from inguinal lymph nodes”. The X-axis can then indicate the two sources of lymph nodes (PBS control mice and rosoxacin treated mice), each with the two types of in vitro treatments (no rosoxacin and yes rosoxacin). The Y-label will also be on the same range (from 0 to 3) for both groups, which facilitates comparisons.
4) References 1 and 3 refer to the same paper
